# *SlATG8f* modulates tomato thermotolerance and fruit quality, correlating with changes in autophagy and heat shock-related genes

Qunmei Cheng[1], Wen Xu[1,2]*, Cen Wen[1], Zhuo He[1,3], Liu Song[1]

**1** College of Agriculture, Guizhou University, Guiyang, China, **2** Engineering Research Center for Protected Vegetable Crops in Higher Learning Institutions of Guizhou Province, Guiyang, China, **3** Agriculture and Rural Bureau of Chishui City, Zunyi, China

* 15185145028@163.com

## Abstract

Plant responses to high-temperature stress involve intricate changes in physiology bio-chemistry, and gene expression. Autophagy, a conserved intracellular degradation system, aids in eliminating damaged cellular components and is crucial for plant growth, development, and stress adaptation. Despite the known functions of many autophagy-related genes under high temperatures, the specific role of *SlATG8f* in tomatoes remains unclear. In this study, *SlATG8f* OE tomato lines were created using the pBWA(V)HS vector. Through quantitative reverse transcription polymerase chain reaction (qRT-PCR), we assessed physiological parameters, expression levels of *ATG8* family members, and heat shock protein-related genes in fruits of wild-type (WT) and *SlATG8f* OE plants at four developmental stages under high-temperature stress. Our findings demonstrate that *SlATG8f* OE enhances the expression of autophagy-related and heat shock protein genes, accelerates fruit ripening, and enhances fruit quality under high-temperature conditions. These results highlight the regulatory function of *SlATG8f* in preserving tomato fruit quality during heat stress and offer a basis for enhancing tomato varieties.

## Introduction

Autophagy, an evolutionarily conserved process in eukaryotes, facilitates the recycling of intracellular components and damaged macromolecules, playing a pivotal role in plant development and stress adaptation [1]. Among the core autophagy-related (ATG) proteins, ATG8 is crucial for autophagosome formation and function. In plants, the ATG8 gene family has significantly expanded from a single ancestral gene in algae to multiple isoforms in higher plants, reflectin functional diversification [2]. Most autophagic receptors and adaptors feature an *ATG8*-interactin motif (AIM/UIM) that enables specific binding to ATG8 proteins [3]. Increasing evidence suggests that autophagy is triggered under various abiotic stresses, such as nutrient deficiency,

**Data availability statement:** ***A/PROS AT ACCEPT: Please follow up with authors for data available at accept***The data supporting the findings of this study will be deposited in [NCBI GenBank] and will be publicly available upon article publication.

**Funding:** This work was supported by the National Natural Science Foundation of China (32260754).

**Competing interests:** The author declares no conflict of interest.

salinity, drought, and extreme temperatures, generally promoting cell survival and preserving cellularhomeostasis [4]. However, the role of autophagy may vary depending on the stress type, tissue, and developmental stage. For instance, in barley, autophagy has been associated with programmed cell death in microspores under cold stress, with its inhibition leading to reduced cell death [5], indicating that autophagy can exhibit either pro-survival or pro-death functions contingent on the context [6].

Heat stress (HS) is a significant limitation on crop productivity and quality, triggering the upregulation of autophagy-related genes and facilitating autophagosome formation [7].The degree of autophagic activation is influenced by the severity and duration of stress, as well as the plant's genotype. Research utilizing Arabidopsis loss-of-function mutants (*ATG5* and *ATG7)* has demonstrated heightened heat susceptibility, leading to more pronounced harm to photosynthetic and membrane systems [8]. Conversely, the overexpression of specific *ATG* genes, such as *MdATG18a* in apple, has been found to enhance thermotolerance by mitigating oxidative damage and safeguarding chloroplast function [9]. Various autophagy genes, including *ATG5, ATG6, ATG12a, ATG18a, ATG8a, ATG8c, ATG8g,* and *ATG8i,* have been implicated in plants' response to high-temperature stress [8,10,11]. Nevertheless, the role of *SlATG8f,* a pivotal member of the tomato *ATG8 f*amily, in high-temperature conditions remains inadequately elucidated.

Tomato (*Solanum lycopersicum* L.) serves as a prominent greenhouse crop and a valuable model species for investigating fruit development and stress responses. HS significantly hindersvarious reproductive processes, including pollen viability, fertilization, and fruit set, resulting in substantial yield reduction and quality deterioration [12,13]. Elevated temperatures above 35°C disrupt crucial physiological and biochemical pathways, negatively impacting sugar metabolism, pigment accumulation, and fruit firmness [14].Moreover, HS disrupts root–shoot signaling and nutrient allocation, further compromising fruit development [15].These adverse effects highlight thenecessity of elucidating the molecular mechanisms that could be manipulated to enhance thermotolerance. Although autophagy in crops has been extensively researched in contexts such as leaf senescence, seed development, and vascular formation [16], its involement in the ripening of fleshy fruits, a process involving substantial alterations in color, texture, and metabolite profiles, remains relatively unexplored.

Autophagy plays a crucial role in fruit maturation, as indicated by recent studies. Transcriptomic analysis of grape berry skin showed increased expression of ATG genes during maturation, suggesting a link between autophagy and fruit senescence [17]. Similar findings were observed in pepper fruit ripening, where autophagy-like structures and the expression of autophagy related genes and proteins were detected [18]. In strawberry, inhibition of autophagy disrupted the ripening process, highlighting its importance in vascular development and senescence [19]. A study in tomato revealed that *SlATG8f* coordinates ethylene signaling and chloroplast turnover to drive fruit ripening [20]. These studies collectively suggest that autophagy serves as a key regulator of metabolic remodeling and quality acquisition during fruit development, potentially playing a protective role under abiotic stresses like high temperatures.

In this study, transgenic tomato lines were developed to overexpress *SlATG8f* and were exposed to sustained high-temperature stress (35°C) during fruit maturation. Expression patterns of ATG8 family members and heat shock protein (HSP) genes were analyzed in WT and transgenic plants at four fruit developmental stages using qRT-PCR. Fruit quality parameters including external color, firmness, sugar-acid ratio, and carotenoid content were assessed. The results indicate that overexpression of *SlATG8f* enhances autophagic activity, upregulates HSP gene expression under heat stress, accelerates fruit ripening, and improves overall fruit quality. These findings provide insights into the role of *SlATG8f* in thermotolerance and fruit development, presenting a valuable genetic tool for developing tomato varieties with enhanced heat tolerance and superior fruit quality.

## Materials and methods

### Plant materials and growth conditions

The study employed the dwarf tomato 'Micro-Tom' from the College of Agriculture at Guizhou University. Seedlings were grown in a soilless substrate consisting of peat, vermiculite, and perlite in a 2:1:1 volume ratio. The specific procedures were as follows: Seeds were surface-sterilized with 0.5% sodium hypochlorite for 15 minutes, rinsed 3−5 times with sterile water, immersed in a 55°C water bath with gentle agitation for 15 minutes, and then incubated in sterile water at 28°C with shaking at 100 rpm for 12 hours. Subsequently, the seeds were placed on moist filter paper and kept in the dark at 28°C for 24 hours to promote germination. Germinated seeds were sown in 72-cell seedling trays. Seedlings were grown under controlled environmental conditions: a 16-h light (25°C)/8-h dark (18°C) photoperiod, 80% relative humidity, and a photosynthetic photon flux density of 250 μmol m$^{-2}$s$^{-1}$. Upon reaching the three-true-leaf stage, seedlings were transplanted into 12 cm diameter pots with the same soilless substrate. Throughout the growth phase, plants were watered with Hoagland's nutrient solution every five days.

### Generation of transgentic tomato lines

The *SlATG8f* OE vector was created by inserting the *SlATG8f* coding sequence into the pBWA(V)HS vector. Healthy tomato cotyledons were inoculated with *Agrobacterium tumefaciens* strain GV3101 carrying the *35S::SlATG8f* construct, leading to the generation of transgenic lines via Agrobacterium-mediated transformation. The upregulation of *SlATG8f* in the transgenic lines was validated using qRT-PCR. Homozygous $T_2$ plants were specifically chosen for subsequent analyses [21].

### Heat stress treatment and sample collection

When the initial fruit truss entered the rapid expansion phase, nine plants of both the wild-type and *SlATG8f*OE (homozygous F2 generation) were moved to a controlled growth chamber for heat treatment at 35°Cday/21°Cnight, while plants grown at 25°Cday/16°Cnight were used as the control [22],Fruit samples were obtained at four developmental stages: mature green, breaker, orange ripe, and red ripe. For each biological replicate within a treatment group, fruits were randomly chosen from a minimum of three different plants and from various locations (e.g., upper, middle) on the truss to reduce positional bias. The pericarp tissues from these fruits were then removed and combined to create a single composite sample for subsequent RNA, biochemical, and quality analyses.

### Phenotypic assessment of fruit development

On Day 0, three plants with synchronously developing fruits were randomly chosen per biological replicate for transfer to control or heat-stress growth chambers. From each plant, 3–5 fruits at the expansion stage were tagged. This process was replicated for three independent biological replicates per treatment. Starting from Day 0, the tagged fruits were observed daily until the first visible color transition. The duration to reach the mature green, breaker, orange ripe, and red

ripe stages was noted. Upon the first signs of transitioning into each stage, fruit samples were promptly collected, frozen in liquid nitrogen, and stored at −80°C for subsequent physiological and gene expression analyses. The transition time to each stage for each biological replicate was calculated as the average across all labeled fruits from the three plants. The final values in the figures represent the means of the three biological replicates. To examine locular gel development, one fruit at various ripening stages was randomly selected from each treatment group and longitudinally sectioned using a laboratory scalpel.

Fruit set rate served as a crucial indicator of reproductive thermotolerance. Upon initiating the treatments (Day 0), the initial inflorescence on each plant received a tag, with all open flowers on that inflorescence being individually labeled. The count of successfully set fruits on each tagged inflorescence was conducted 15 days post-flowering. A fruit was deemed successfully set if it remained without abscission and attained a diameter exceeding 5 mm. The fruit set rate for each plant was determined by the formula: (Number of successfully set fruits/Total number of labeled flowers)×100% Each treatment comprised three plants as one biological replicate, with three distinct biological replicates analyzed per treatment. The ultimate fruit set rate for each experimental cohort was denoted as the average of the three biological replicates.

## Analysis of fruit quality parameters

For each treatment group, five fruits were randomly selected at various ripening stages. Fruit firmness was assessed at the equatorial region using a GY-B fruit hardness tester, with three replicate measurements per fruit. Soluble solid content (SSC) and titratable acidity (TA) were determined using a PAL-BX/ACID1 refractometer (ATAGO Co., Japan). Carotenoid content was quantified following the method outlined in the literature [23], with slight modifications. In summary, three fruits were combined to create a composite sample, finely ground under liquid nitrogen, and 0.2 g of the homogenized powder was extracted with a 10 mL hexane-acetone mixture (3:2, v/v). The mixture was vortexed for 5 minutes, then left to incubate in darkness on ice overnight. Following incubation, samples were centrifuged at 5000 rpm for 5 minutes. A suitable volume of the supernatant was transferred to a 1 mL cuvette, and the absorbance was measured at 470 nm, 649 nm, and 665 nm using a spectrophotometer to determine the carotenoid content.

## Measurement of antioxidant enzyme activities and Malondialdehyde (MDA) content

Physiological indicators such as MDA and antioxidant enzyme activity were analyzed using the UV-5500 UV-Vis spectrophotometer (Shanghai DaPu Instrument China).The MDA content was determined using the thiobarbituric acid (TBA) method with slight modifications, following the protocol by [24]. Three randomly selected fruits from each treatment group were homogenized after weighing0.1g and grinding in liquid nitrogen. The homogenate was centrifuged at 3000 rpm for 10 minutes after adding 1 mL of 5% TAC. The supernatant (2 mL) was slowly aspirated with a pipette, mixed with 2 mLof 0.67%TBA, and boiled in a water bath at 95°C for 15 minutes. After cooling on ice, the mixture was centrifuged, and the absorbance values at 450 nm, 532 nm, and 600 nm were measured in the supernatant. Distilled water with 0.67%TBA served as the blank control.

CAT, POD, and SOD enzyme activities were determined according to the references provided [25,26]For enzyme extraction, 0.1 g of leaf tissue was added to pre-chilled phosphate buffer solution containing 50 mM sodium phosphate (pH 7.8), 0.2 mM EDTA, and 2% (w/v) PVP, homogenized, and centrifuged at 12000 rpm for 20 minutes. The resulting supernatant was used for enzyme activity assays.

CAT activity was determined by mixing 100 μL of enzyme solution with 1700 μL of 25 mM PB buffer (pH 7.0, containing 0.1 mM EDTA) and 200 μL of 10mM $H_2O_2$. After incubation at 25°C for a few minutes, the reaction was stopped, and the absorbance at 240 nm was measured before and after 1 minute. One unit of enzyme activity was defined as the amount of enzym required to decrease the absorbance at 240 nm by 0.01 per minute per gram of fresh tissue.

SOD activity was determined by incubating 50 μL of enzyme solution with 3 mL of 50 mM PB buffer (pH 7.8, containing 15 mM methionine, 65 μM NBT, 2.0 μM riboflavin, and 10 mM EDTA) under light at 25°C for 5 minutes. The reaction was

stopped in the dark, and the absorbance at 560 nm was immediately measured. One unit of enzyme activity was defined as the amount of enzyme required to inhibit 50% of the NBT photoreduction reaction per gram of fresh tissue.

POD activity was determined by mixing 100 µL of enzyme solution with a mixture of 28 µL of 50 mM guaiacol and 34 µL of 30% $H_2O_2$ in 100 mM phosphate buffer (pH 7.0). The absorbance at 470 nm was measured before and after 1–2 minutes, and enzyme activity was expressed as the increase in absorbance of 0.01 per minute per gram of fresh tissue. The enzyme solution was used as a control after heat inactivation (5–10 minutes).

### RNA extraction and gene expression analysis (qRT-PCR)

Total RNA was extracted from tomato fruit tissues using the RNAprep Pure Plant Kit (TIANGEN, Cat#DP441) designed for polysaccharide- and polyphenol-rich samples following the manufacturer's instructions. First-strand cDNA was synthesized using the StarScript III All-in-one First-Strand cDNA Synthesis Kit (with gDNA remover, GenStar, Beijing, China). Gene-specific primers were designed using Premier 5.0 software and obtained from a commercial supplier. The gene names and primer sequences can be found in Supplementary S1 and S2 Tables. Quantitative real-time PCR was conducted using 2× RealStar Fast SYBR qPCR Mix (GenStar, Beijing, China). Each sample included three biological replicates and five technical replicates. The Actin gene (*Solyc03g078400* ) was chosen as the reference gene based on previous studies [21]. The qRT-PCR reaction system and cycling program followed the manufacturer's instructions for the 2× RealStar Fast SYBR qPCR Mix. Relative gene expression levels were determined using the $2^{-\Delta\Delta CT}$ method.

### Statistical analysis

The experiment was conducted following a completely randomized design with independentbiological replicates, where each treatment group consisted of three replicates. Random sampling was conducted within each group. Data organization was done using Microsoft Excel 2019 and analysis was performed using SPSS Statistics 23.0. Statistical significance was assessed using Duncan's multiple range test, with significance determined at $P < 0.05$, with differences indicated by distinct lowercase letters. Figures were created using Origin 2022, and all data were reported as mean ± standard deviation.

## Results

### Overexpression of *SlATG8f* improves locular gel formation and provides morphological resilience to heat stress

To assess the impact of *SlATG8f* OE plants on tomato fruit development under heat stress, fruit phenotypes were compared between treatment groups Fig 1. No significant differences were observed between WT and *SlATG8f* OE fruits under control conditions. However, heat stress led to fruit deformation in both genotypes, with more pronounced malformations in *SlATG8f* OE lines. Anatomical analysis revealed that under normal conditions, *SlATG8f* OE fruits exhibited accelerated and earlier completion of locular gel formation compared to WT fruits. When exposed to heat stress, gel development was delayed in both WT and *SlATG8f* OE fruits, although *SlATG8f* OE plants demonstrated higher tolerance than WT.

### Overexpression of *SlATG8f* improves fruit set under high-temperature stress

To investigate the impact of *SlATG8f* OE plants on fruit color transition and fruit set under high-temperature conditions, we monitored the timing of four fruit developmental stages (Green ripening, Colour breaking, Orange ripening, and Red ripening) and assessed fruit set rates in both heat-treated and control plants Fig 2. In control conditions, all developmental stages occurred earlier in WT plants compared to *SlATG8f* OE plants. Conversely, high-temperature stress delayed the progression of these stages in WT plants but accelerated maturation in *SlATG8f* OE fruits (Fig 2A). While WT plants showed a higher fruit set rate than *SlATG8f* OE plants under normal conditions, heat stress reduced fruit set significantly in WT plants. In contrast, *Sl-ATG8f* OE plants maintained a consistent fruit set rate under heat stress (Fig 2B). Although

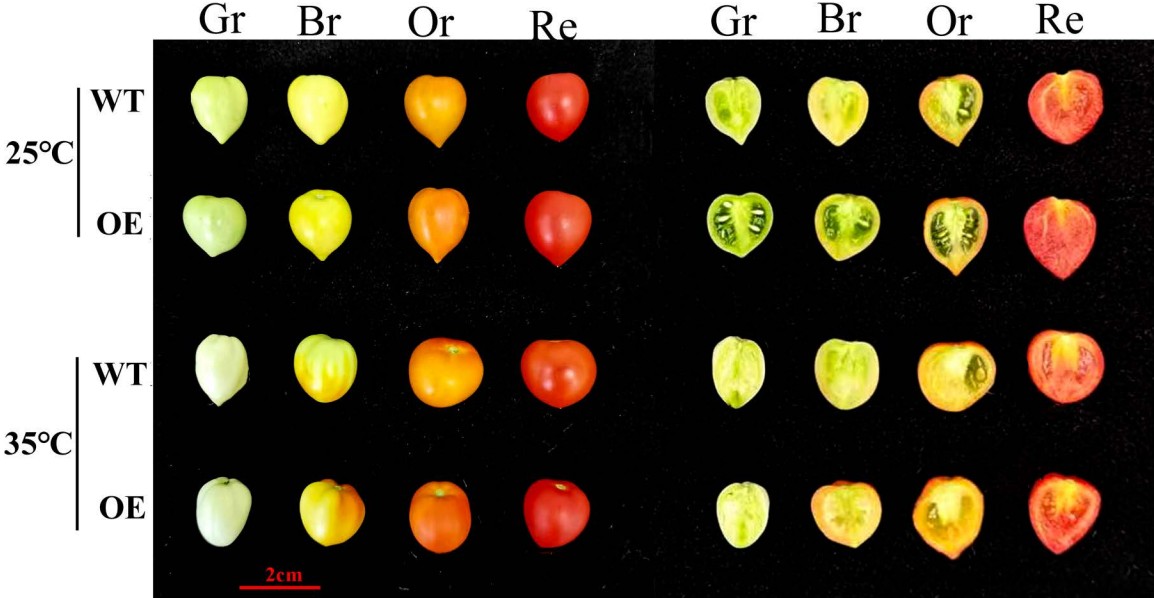

**Fig 1. Impact of ambient control and high temperature treatments on phenotypic changes in tomato f-ruit.** The temperature of 25°C represents the control group, while 35°C represents the high-temperature treatment.WT (Wild Type), OE (*SlATG8f* OE plants), Gr(Green ripening), Br(Colour break-ing),Or (Orange ripening), Re (Red ripening),scale bar (2 cm).

no significant difference in fruit set was observed between WT and *SlATG8f*OE lines under high temperature, the trans-genic plants effectively mitigated the heat-induced decline in fruit set. These findings indicate that *SlATG8f* OE enhances thermotolerance by expediting fruit ripening and preserving fruit set under high-temperature stress.

## The modulation of *SlATG8f* OE affects tomato fruit quality under heat stress

To assess the impact of *SlATG8f* OE plants on tomato fruit quality during high-temperature stress, essential quality param-eters were assessed in WT and *SlATG8f* OE fruits under control and stress conditions Fig 3.

Fruit firmness was generally higher in *SlATG8f* OE fruits compared to WT, except at the green ripe stage, where WT fruits exhibited slightly greater firmness. Elevated temperatures further enhanced firmness in both genotypes (Fig 3A). Fruits solid content remained consistent between genotypes across treatments; however, heat stress led to increased SSC levels at the breaker and red ripe stages while decreasing it at the green and orange ripe stages (Fig 3B).

The titratable acid (TA) content was consistently higher in *SlATG8f* OE fruits compared to WT fruits under all conditions and was slightly increased by high temperatures across developmental stages (Fig 3C). The solid-acid ratio was higher in WT fruits and decreased under heat stress in both lines (Fig 3D).

Carotenoid content was higher in WT fruits under control conditions. However, heat stress led to a reduction in carot-enoid accumulation in WT fruits at the orange and red ripe stages, resulting in lower levels compared to *SlATG8f* OE fruits under stress (Fig 3E).

In general, *SlATG8f* OE fruits displayed higher firmness, soluble solids content (SSC), and TA under high-temperature stress in comparison to WT fruits.

## Effect of *SlATG8f OE* plants on enzyme activities in tomato fruit under high temperature

To assess the influence of *SlATG8f* OE plants on antioxidant enzyme activities under high-temperature stress, key bio-chemical markers were examined in fruits from WT and *SlATG8f* OE plants under normal and stress conditions Fig 4.

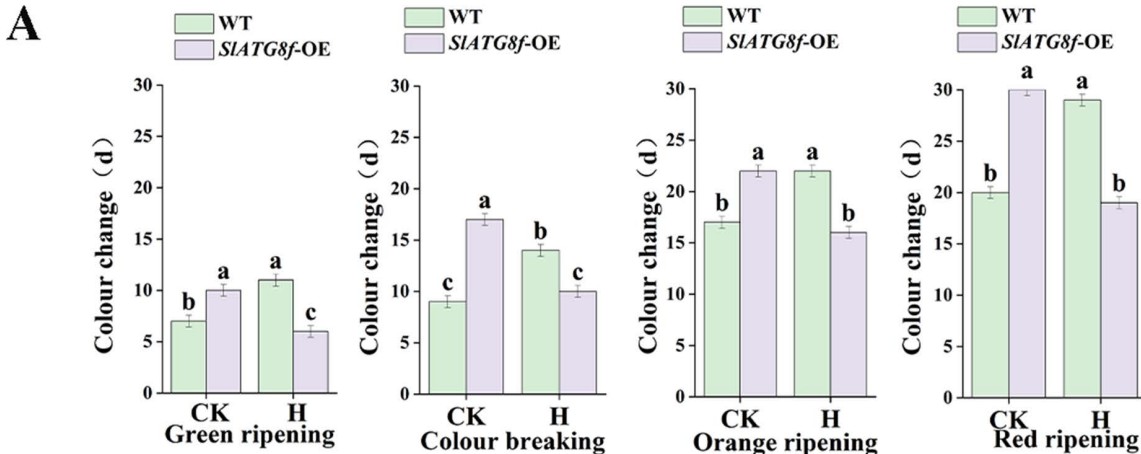

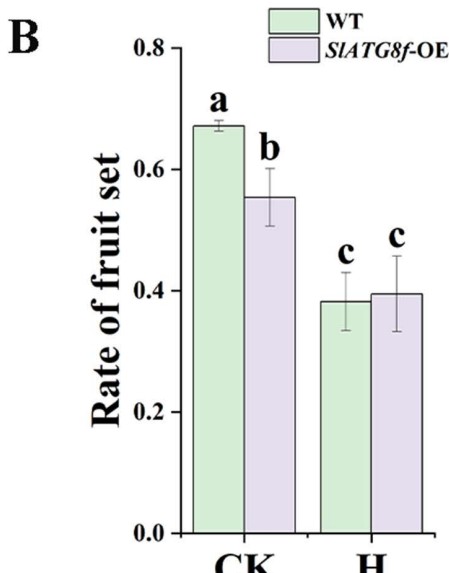

**Fig 2. Effects of ambient control and high temperature treatments on fruit set in tomato plants.** A. The number of days required for fruit ripening stages, from left to right, are the green ripening period, breaker stage, orange ripening period, and red ripening period. B. Result rates of the wild type and *SlATG8f* OE plants. CK (control), H (high-temperature treatment), WT (Wild Type), *SlATG8f* -OE (*SlATG8f* OE plants), ($P < 0.05$).

Elevated temperatures increased catalase (CAT) activity, consistently higher in WT fruits compared to *SlATG8f* OE fruits across growth stages (Fig 4A). Conversely, peroxidase (POD) ac-tivity was generally higher in *SlATG8f* OE fruits than in WT fruits. Heat stress amplified POD activity at orange and red ripe stages but suppressed it at green and breaker stages (Fig 4B).

Malondialdehyde (MDA) content was lower in *SlATG8f* OE fruits compared to WT fruits, except at the orange ripe stage. Under high temperature, MDA content rose in WT fruits at the orange ripe stage but declined at the other stages (Fig 4C).

In normal conditions, superoxide dismutase (SOD) activity was greater in *SlATG8f* OE fruits than in WT fruits. However, heat stress increased SOD activity in WT fruits while decreasing-g it in *SlATG8f* OE fruits (Fig 4D).

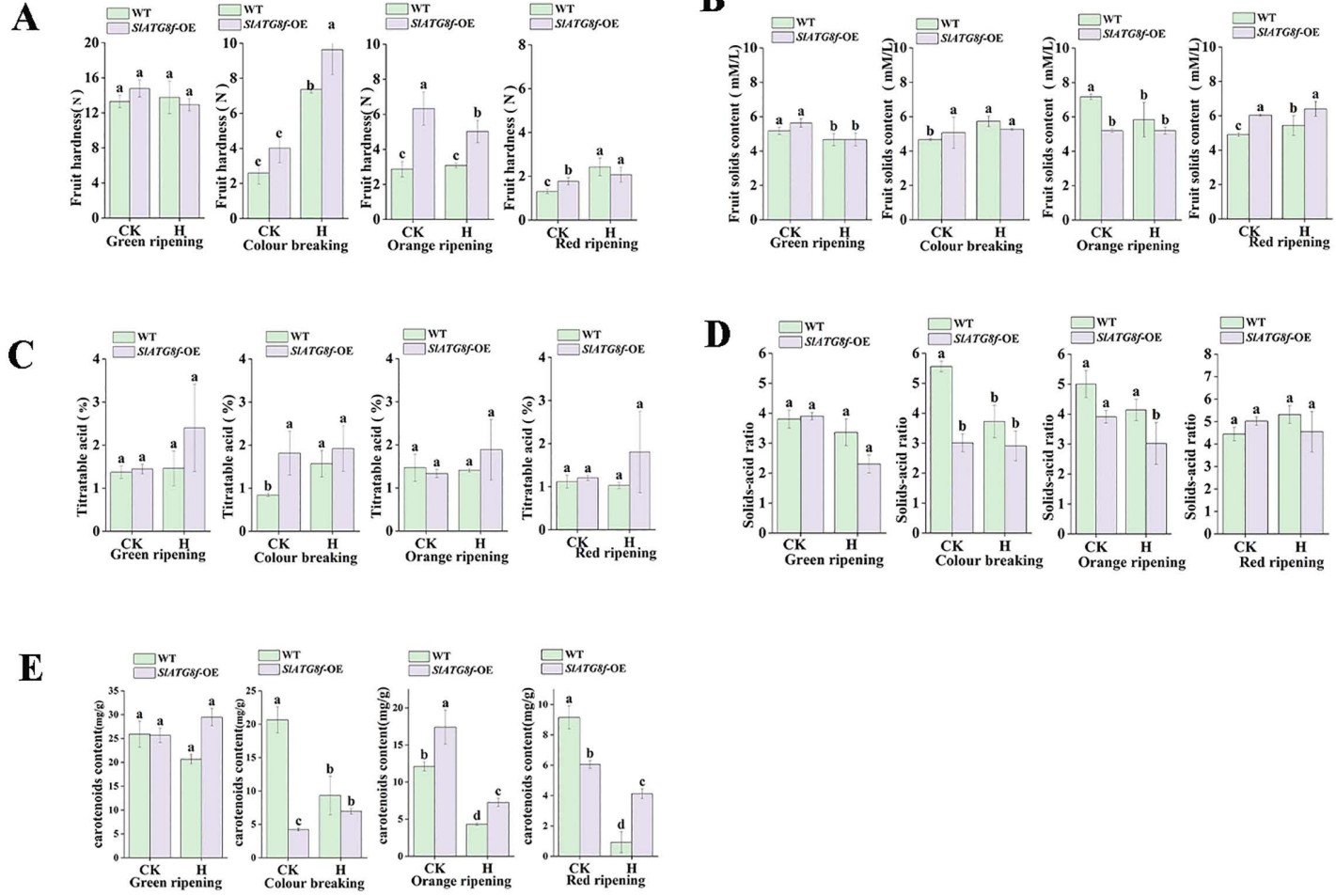

**Fig 3. Modifications in fruit quality indicators of tomato plants under ambient control versus high temperature conditions.** A. Fruit firmness. B. Fruit solids content. C. Titratable acidity. D. Firmness-to-acidity ratio. E. Carotenoid content. CK (control), H (high-temperature treatment), WT (Wild Type), *SlATG8f* -OE (*SlATG8f* OE plants), A (fruit hardness), B (soluble solids content), C (titratable acid), D (solids-acid ratio), E (carotenoids content), ($P < 0.05$).

## The expression of heat-shock protein response genes in tomato fruit is modulated by heat stress

Heat stress response gene expression was examined in WT and *SlATG8f* OE fruits under control and high-temperature conditions Fig 5. In control conditions, *SlHSFA2* expression was highest in *SlATG8f* -OE fruits at the breaker stage and remained stable under heat stress (Fig 5E). Conversely, *SlHSFA3* (Fig 5F), *SlMYB21* (Fig 5G), and *SlMYB26* (Fig 5H) displayed low overall expression (Fig 5F), with higher levels of *SlHSFA3* and *SlMYB26* in WT compared to *SlATG8f* OE fruits. *SlHSFA2* exhibited higher expression in WT, while *SlMYB21* was elevated in *SlATG8f* OE fruits. Heat stress induced upregulation of *SlHSP20* (Fig 5A), *SlHSP21* (Fig 5B), and *SlHSP70* (Fig 5C) in both genotypes, with stronger induction in WT. Notably, *SlHSP90* (Fig 5D) expression peaked in WT fruits at the green ripe stage under control condit-ions and at the breaker stage under heat stress, significantly surpassing levels in *SlATG8f* OE fruits across all developmental stages.

## Impact of heat stress on *ATG8* family gene expression in tomato fruit

This study investigated the impact of high-temperature stress on autophagic activity in tomato fruit by assessing the expression of *ATG8* family genes using qRT-PCR Fig 6. The fin-dings revealed that *SlATG8a* expression remained

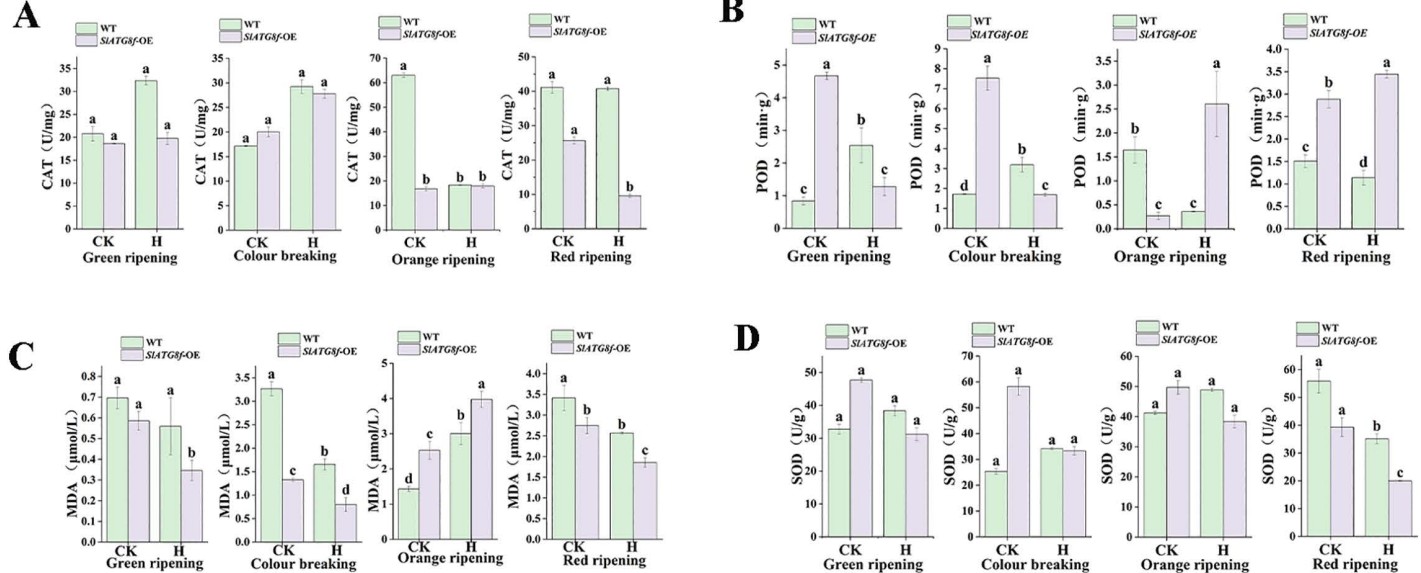

**Fig 4. Assessment of antioxidant enzyme activities in tomato fruits exposed to ambient control and high temperature conditions.** A. Catalase (CAT) activity. B. Peroxidase(POD) activity. C. Malondialdehyde (MDA) content. D. Superoxide dismutase (SOD) activity. CK (control), H (high-temperature treatment), WT (Wild Type), *SlATG8f*-OE (*SlATG8f* OE plants)., ($P < 0.05$).

consistent in both WT and *SlATG8f* OE fruit-s, unaffected by temperature (Fig 6A). *SlATG8b* expression displayed a unimodal trend throughout fruit development stages (Fig 6B). In WT fruits, *SlATG8c* exhibited its highest expression under normal conditions, maintaining relatively stable levels post-breaker stage (Fig 6C),. Conv-ersely, *SlATG8d* peaked at the breaker stage in *SlATG8f* OE fruits under both conditions, while consistently low in WT fruits (Fig 6D). Overall, *SlATG8d* levels were elevated in *SlATG8f* OE fruits compared to WT, showing an initial rise followed by a decline. *SlATG8e* expression was highest in WT fruits at the green ripe stage under normal conditions, remaining steady thereafter (Fig 6E). *SlATG8f* expression was higher in*SlATG8f* OE fruits than in WT at the same stage (Fig 6F),. Meanwhile, *SlATG8g* expression was nearly negligible at the green and breaker stages in WT fruits under normal conditions, generally demonstrating an initial increase followed by a decrease throughout development (Fig 6G).

## Discussion

Fruit development is a critical stage in plant reproduction, involving processes such as cell division, expansion, and metabolic changes. In tomatoes, ripening is characterized by color tra-nsition, tissue softening, and the accumulation of various metabolites [27,28], Autophagy, a conserved degradation pathway in eukaryotes, plays a role in fruit metabolism and nutrient recycle-ng during development, although its function is not well-explored [21]. While autophagy has been extensively studied in vegetative tissues of various crop species [29], its role in fruits under abiotic stress conditions remains poorly understood. Recent evidence suggests that autophagy related genes are upregulated during late ripening stages in grapes and are crucial for quality formation in chili and strawberry fruits [18,19,17], Our findings support these reports by showin-g that *SlATG8f* OE enhances locular gel liquefaction and tissue integrity under normal conditions. Under high-temperature stress, both WT and *SlATG8f* OE fruits exhibited morphological de-formities and delayed gel development, but the transgenic lines maintained superior firmness, s-oluble solid content, and titratable acidity. Importantly, *SlATG8f* OE fruits retained higher carotenoid levels under heat stress compared to WT fruits, which experienced a significant reduction, highlighting the protective role of *SlATG8f* in mitigating heat-induced damage to bioactive compounds. This sensitivity to heat-induced carotenoid reduction aligns with previous observations in tomatoes subjected to prolonged high-temperature treatment [30].

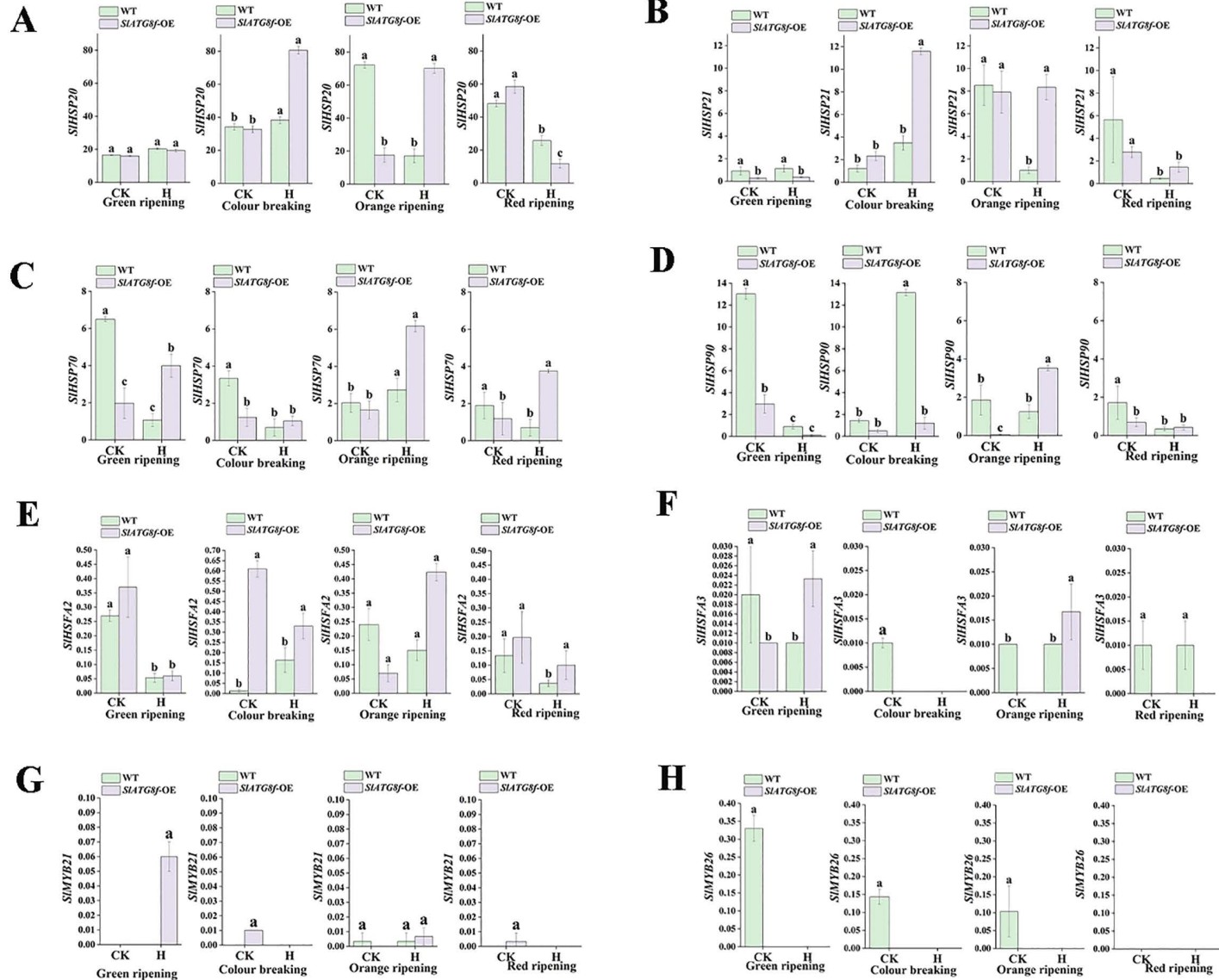

**Fig 5. Differential expression analysis was conducted on heat stress protein response genes in tomato plants subjected to both ambient control and high temperature treatment conditions.** A. Relative expression level of *SlHSP20*. B. Relative expression level of *SlHSP21*. C. Relative expression level of *SlHSP70*. D. Relative expression level of *SlHSP90*. E. Relative expression level of *SlHSFA2*. F. Relative expression level of *SlHSFA*3. G. Relative expression level of *SlMYB21*. H. Relative expression level of *SlMYB26*. CK (control), H (high-temperature treatment), WT (Wild Type), *SlATG8f* -OE (*SlATG8f* -OE plants), ($P < 0.05$).

Autophagy is a crucial response activated during heat stress to eliminate damaged proteins and organelles [31]. Mutations in core autophagy genes, such as *ATG5* and *ATG7*, have been found to increase thermosensitivity in Arabidopsis and tomato [8,32], underscoring the significance of autophagy in thermotolerance. The heat stress response involves intricate regulatory mechanisms, including the unfolded protein response and chaperone signaling pathways [31,33]. Our findings indicate that the overexpression of *SlATG8f* boosts the expression of multiple *ATG8* family members and heat shock proteins (HSPs) under high-temperature conditions. While certa-in HSPs, like *SlHSP90*, showed higher expression

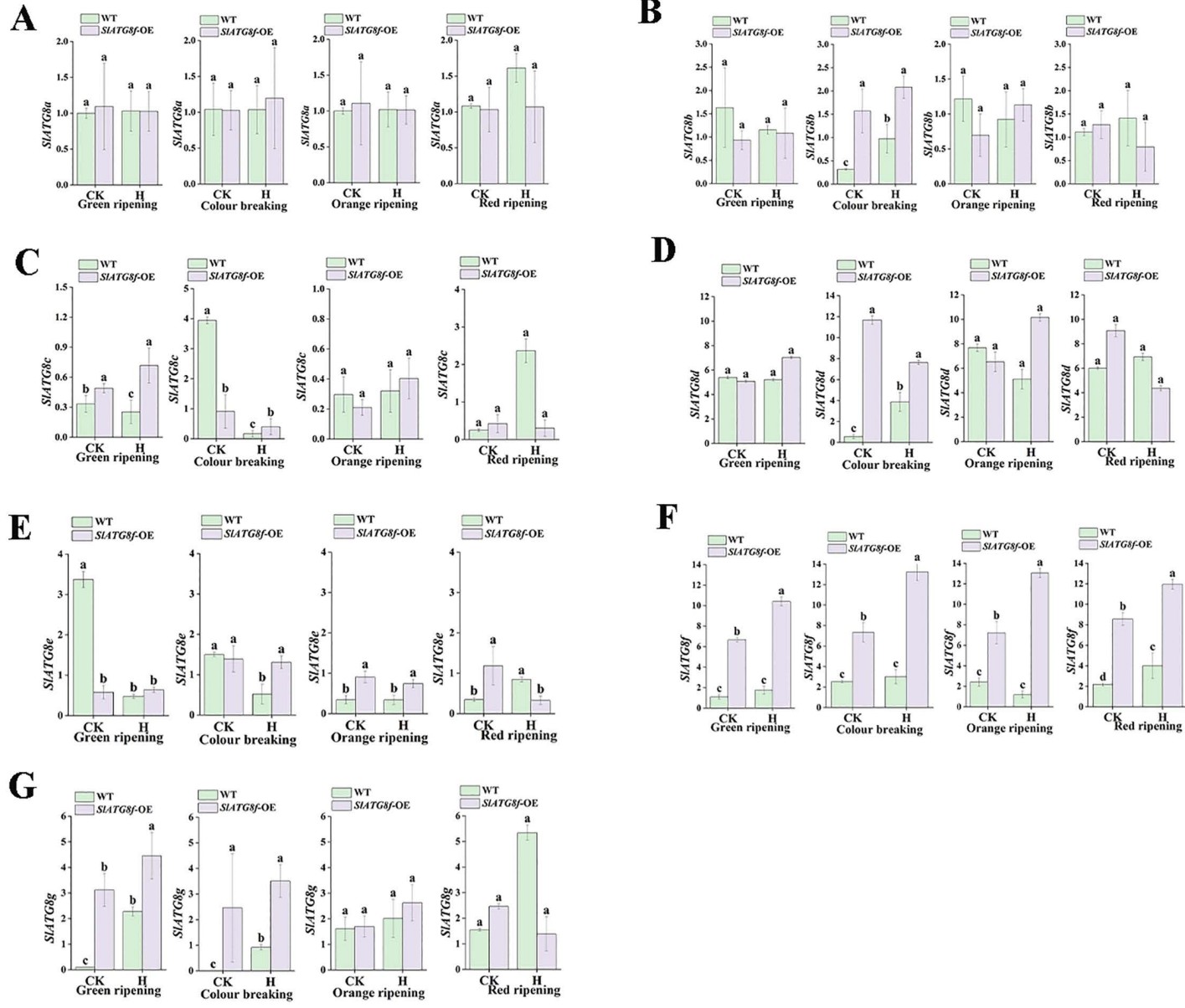

**Fig 6. Investigation of the varied expression of *SlATG8s* in tomato fruit subjected to ambient control and high-temperature conditions.** A. Relative expression level of *SlATG8a*. B. Relative expression level of *SlATG8b*. C. Relative expression level of *SlATG8c*. D. Relative expression level of *SlATG8d*. E. Relative expression level of *SlATG8e*. F. Relative expression level of *SlATG8f*. G. Relative expression level of *SlATG8g*. CK (control), H (high-temperature treatment), WT (Wild Type), *SlATG8f*-OE (*SlATG8f* OE plants), ($P < 0.05$).

in WT fruits, most HSPs and autophagy rel-ated genes were notably upregulated in *SlATG8f* OE fruits, suggesting a syn-chronized activation of protein quality control pathways facilitated by *SlATG8f* during heat stress.

However, there are still some limitations in this study, which also point the way for future indepth research. Primarily, Our conclusions are primarily drawn from an *SlATG8f* OE model. While overexpression offers valuable insights, generat-ing and analyzing loss-of-function mutan-ts, potentially through CRISPR/Cas9, is crucial to fully validate *SlATG8fs* role in thermotolera-nce. Just as previous studies [20] utilized VIGS technology to illustrate *SlATG8f's* involvement in ethylene

signaling and chloroplast turnover during ripening under normal conditions, obtaining reverse genetic evidence, such as through CRISPR/Cas9 mutagenesis, would definitively establish the importance of *SlATG8f* in thermotolerance. Furthermore, previous research [34]and [35]demonstrated the role of autophagy in aging and stress response using *ATG* mutants in Arabidopsis thaliana and tomato, respectively, serving as a reference for constructing stable mutants in future studies.

Secondly, The study utilized qRT-PCR data to propose increased autophagy activity. However, to confirm autophagy activation, detection of *ATG8* lipidation (*ATG8*-PE), a key feature of autophagosome formation – is recommended.Therefore, Western blot analysis of lipid-modified *SlATG8f* protein is essential as the preferred method to validate enhanced autophagic flux, as advised by established autophagy monitoring guidelines [36]. Moreover, Western blot analysis of*ATG8* lipidation is commonly used in plant autophagy research [37], Future investigations focusing on the dynamic alterations in *SlATG8f* protein levels and lipidation would enhance the robustness of our findings.

Finally, The mechanism underlying the regulation of HSP gene expression by *SlATG8f* remains ambiguous. It is unclear whether *SlATG8f* directly interacts with heat shock transcription factors like *SlHSFA2* or *SlHSFA3*, or if it indirectly boosts the heat shock response through selective autophagy of negative regulators of the HSP pathway. Techniques such as yeast two-hy-brid (Y2H), luciferase complementation assay (LCI), or co-immunoprecipitation (Co-IP) could elucidate these interactions. Prior studies offer methodological insights; for instance, the documented interaction between plant *ATG8* and specific target protein [38]and the regulation of leaf aging in soybean by *GmATG8s* through interaction with *GmHSP90* [39].These findings serve as a valuable reference for exploring S*lATG8f*–*HSP* interactions in tomato.

In summary, this investigation highlights the positive role of *SlATG8f* in enhancing thermotolerance in tomato fruit under high-temperature conditions. The findings offer a valuable genetic asset and theoretical groundwork for enhancing crop resilience to stress and fruit quality through genetic modification. Subsequent studies should concentrate on developing stable loss-of function mutants, directly assessing autophagic flux at the protein level, and clarifying the interaction between *SlATG8f* and heat shock signaling components to comprehensively elucidate the molecular pathways governed by this gene.

## Conclusions

In this study, WT and *SlATG8f* OE tomato plants were exposed to high-temperature stress (35°C), with 25°C as the control. Comparative analysis showed that *SlATG8f* OE improved thermotolerance during fruit development, enhancing phenotypic traits, accelerating fruit set, color transition, and maturation. Additionally, *SlATG8f* OE upregulated the expression of various ATG8 family members and heat shock protein-related genes under high-temperature conditions. The results indicate that the enhancement of thermotolerance and fruit quality under heat stress by *SlATG8f* OE is associated with heightened autophagic activity and the activation of heat stress response mechanisms. This study highlights the crucial role of *SlATG8f* in regulating tomato fruit development under abiotic-stress, offering a theoretical basis for enhancing tomato quality and stress resilience in breeding programs.

## Supporting information

**S1 Table. Primer Sequences and Reference Genes.**
(PDF)

**S2 Table. The qRT-PCR reaction system and its corresponding reaction program.**
(PDF)

## Acknowledgments

We gratefully acknowledge the financial support for this research from the National Natural Science Foundation of China (NSFC). Additionally, we extend our appreciation to the editors and reviewers for their insightful feedback and suggestions, which have significantly enhanced the quality of this manuscript.

## Author contributions

**Conceptualization:** Qunmei Cheng, Wen Xu.

**Data curation:** Qunmei Cheng.

**Formal analysis:** Qunmei Cheng, Cen Wen.

**Funding acquisition:** Wen Xu.

**Investigation:** Wen Xu, Zhuo He.

**Methodology:** Qunmei Cheng.

**Project administration:** Wen Xu.

**Resources:** Wen Xu.

**Software:** Qunmei Cheng, Cen Wen, Liu Song.

**Supervision:** Wen Xu.

**Validation:** Qunmei Cheng, Cen Wen, Zhuo He.

**Visualization:** Qunmei Cheng, Wen Xu, Liu Song.

**Writing – original draft:** Qunmei Cheng.

**Writing – review & editing:** Wen Xu.

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
