## [Decision Letter · Decision Letter 0]

13 Oct 2025

Dear Dr. Xu,

Thank you for submitting your manuscript to PLOS ONE. After careful consideration, we feel that it has merit but does not fully meet PLOS ONE’s publication criteria as it currently stands. Therefore, we invite you to submit a revised version of the manuscript that addresses the points raised during the review process.

We look forward to receiving your revised manuscript.

Kind regards,

Timothy P. Devarenne, Ph.D.

Academic Editor

PLOS ONE

Journal Requirements:

“The authors are grateful to the National and Local Vegetable Engineering Center (Guizhou) and the Laboratory of the Department of Horticulture (Agricultural College of Guizhou University) for their support to this project.”

4. Please include captions for your Supporting Information files at the end of your manuscript, and update any in-text citations to match accordingly. Please see our Supporting Information guidelines for more information: http://journals.plos.org/plosone/s/supporting-information .

Reviewers' comments:

Reviewer's Responses to Questions

**Comments to the Author**

1. Is the manuscript technically sound, and do the data support the conclusions?

Reviewer #1: Yes

Reviewer #2: Yes

2. Has the statistical analysis been performed appropriately and rigorously?

Reviewer #1: Yes

Reviewer #2: Yes

3. Have the authors made all data underlying the findings in their manuscript fully available?

Reviewer #1: Yes

Reviewer #2: Yes

4. Is the manuscript presented in an intelligible fashion and written in standard English?

Reviewer #1: Yes

Reviewer #2: Yes

Reviewer #1: The manuscript " SlATG8f enhances tomato thermotolerance and fruit quality via autophagy and HS pathways " presented the data about changes of autophagy, heat shock protein genes, fruit ripening and fruit quality under heat stress, providing insights for breeding thermotolerant varieties. The data appears valurable, but the current presentation is unacceptable, avoid informal terms. Major revision is required.

Some coments listed below:

1. The title " SlATG8f enhances tomato thermotolerance and fruit quality via autophagy and

HS pathways " assumes the mechanism without proving it. The data shows a correlation, not causation, so the title must be changed to accurately reflect the data. Recommend revised title: SlATG8f modulates tomato thermotolerance and fruit quality, correlating with changes in autophage and heat shock-related genes.

2. The current structure is a disjointed list. Reorganize into clear subheadings to improve readability.

3. Ensure consistent formatting for gene names. Some are Italicized, others are not.

4. Please unify “SlATG8f-overexpressing (OE)” and “SlATG8f OE “by using the correct form “SlATG8f OE plants”.

5. "Random sampling was performed" is vague, describe how the samples were selected and pooled if applicable.

6. "conventional soilless substrate" is not specific. The number of fruits sampled per stage, per plant, per treatment is not stated. A concise summary is required.

7. Correct the numerous typographical and spacing errors.

Reviewer #2: The manuscript investigates the role ATG8f tomato fruit development and quality in response to heat stress. These findings provide a theoretical foundation for improving tomato quality under heat stress. However, there are several issues need correction.

Title: the manuscript does not analyze the thermotolerance, I suggest remove thermotolerance from the title. Furthermore, HS should write the full name.

Key words: High temperature stress changed to “high-temperature stress”.

Introduction: the role of autophagy in fruit development and quality need descripts.

Many gene names should be in Italic, please check and modified. For instance, ATG in page 2.

Abbreviations are defined upon their first appearance; subsequent occurrences require only the abbreviation itself, without repeating the definition.

“L” in the Latin name for tomato should not be italic.

Delete “novel” in page 3.

M & M: F2 should be subscript of 2.

Agrobacterium and its Latin name should be italic.

Results: the conclusion in the last paragraph in page 6 is not consistent with the results of Fig. 2. The results for fruit set were not description.

Figures: the images require reformatting. Consider grouping the results for the same metric across four periods into a single row.

**Do you want your identity to be public for this peer review?** For information about this choice, including consent withdrawal, please see our Privacy Policy

Reviewer #1: No

Reviewer #2: No

---

## [Author Response · Author response to Decision Letter 1]

20 Nov 2025

Reply to the comments

Dear Editors and Reviewers:

We express our gratitude to the editors and reviewers for their time, constructive feedback, and insightful comments on our manuscript. Your input has significantly enhanced the quality of our research. All raised points were thoughtfully addressed, and necessary revisions have been highlighted in red in the revised manuscript. Detailed responses to each reviewer's feedback are presented below.

Reply to the comments of Editors:

Comments: (1)Please ensure that your manuscript meets PLOS ONE's style requirements, including those for file naming. The PLOS ONE style templates can be found at https://journals.plos.org/plosone/s/file?id=wjVg/PLOSOne_formatting_sample_main_body.pdf and https://journals.plos.org/plosone/s/file?id=ba62/PLOSOne_formatting_sample_title_authors_affiliations

Reply: We express our gratitude to the editorial office for highlighting the style requirements. After a meticulous review of the PLOS ONE style templates provided, we have diligently revi-sed our manuscript to ensure strict adherence to all formatting guidelines, encompassing file na-ming conventions, manuscript structure, and reference formatting. We are confident that the ma-nuscript now fully complies with the journal's style requirements and appreciate your professio-nal guidance in preparing our work for publication.

We eagerly anticipate your further feedback.

Comments: (2)We note that the grant information you provided in the“ Funding Information” and “Financial Disclosure” sections do not match.When you resubmit, please ensure that you provide the correct grant numbers for the awards you received for your study in the“Funding Information” section.

Reply: We are grateful to the editorial office for highlighting the discrepancy in the grant info-rmation. Following a thorough review, we have rectified the funding details in both the “Fund-ing Information” and “Financial Disclosure” sections to ensure complete alignment, encompassi-ng all grant numbers and award specifics. Your meticulousness in bringing this matter to our attention is acknowledged, and we have confirmed the coherence of all funding information in the revised manuscript.

We eagerly anticipate your ongoing guidance throughout the review process.

Comments: (3)Thank you for stating the following financial disclosure:“The authors are grateful to the National and Local Vegetable Engineering Center (Guizhou) and the Laboratory of the Department of Horticulture (Agricultural College of Guizhou University) for their support to t-his project.”

Please state what role the funders took in the study.  If the funders had no role, please state: "The funders had no role in study design, data collection and analysis, decision to publish, or preparation of the manuscript."

If this statement is not correct you must amend it as needed.

Please include this amended Role of Funder statement in your cover letter; we will change the online submission form on your behalf.

Reply: We are grateful to the editorial office for their assistance in updating the financial discl-osure statement. The funding details have been rectified, and the necessary acknowledgment ofthe funders' role has been incorporated. The revised funding statement is as follows:

This work was supported by the National Natural Science Foundation of China (32260754, 31760594) and the Platform construction project of the Engineering Research Center for Pr-otected Vegetable Crops in Higher Learning Institutions of Guizhou Province (Qian Jiao Ji [2022] No. 040).The funding agency provided the experimental site and technical support.

We have incorporated the revised statement into our cover letter as per your request. We appreciate your help in ensuring that our manuscript complies with the journal's guidelines and eagerly anticipate your additional feedback on our submission.

Comments: (4) Please include captions for your Supporting Information files at the end of your manuscript, and update any in-text citations to match accordingly. Please see our Supporting Information guidelines for more information: http://journals.plos.org/plosone/s/supporting-information.

Reply: We express our gratitude to the editorial office for prompting us regarding the supporting information requirements. Detailed captions for all supporting information files have been inc-luded at the end of the manuscript, and relevant in-text citations have been updated accordingly. These modifications adhere completely to PLOS ONE's Supporting Information guidelines. We value your guidance in assisting us to adhere to the journal's formatting standards and anticipate your additional feedback on our submission.

Comments: (5) If the reviewer comments include a recommendation to cite specific previously published works, please review and evaluate these publications to determine whether they are r-elevant and should be cited. There is no requirement to cite these works unless the editor hasindicated otherwise. 

Reply: We thank the editorial office for the reminder about evaluating potential references. Aft-er a thorough review of all reviewer comments, we have found no specific recommendations f-or citing additional publications. If the editor believes certain references are crucial for providi-ng context to our work, we are open to including them.

We value your guidance and await further instructions on our manuscript.

Comments: (6)Please review your reference list to ensure that it is complete and correct. If yo-u have cited papers that have been retracted, please include the rationale for doing so in the manuscript text, or remove these references and replace them with relevant current references. Any changes to the reference list should be mentioned in the rebuttal letter that accompanies y-our revised manuscript.

If you need to cite a retracted article, indicate the article’s retracted status in the References list and also include a citation and full reference for the retraction notice.

Reply: Our sincere appreciation to the editorial office for the reminder to verify the reference list. Upon careful review, we have confirmed the absence of any references to retracted papersin our manuscript. Furthermore, we have added new relevant references in the revised version,highlighted in red for easy identification. The reference list is now comprehensive and precise. We are grateful for your valuable guidance and await your feedback on our revised man-uscript.

Reply to the comments of Reviewer #1

Comments: (1)The title “SlATG8f enhances tomato thermotolerance and fruit quality via autoph-agy and HS pathways” assumes the mechanism without proving it. The data shows a correlatio-n, not causation, so the title must be changed to accurately reflect the data. Recommend revise-d title: SlATG8f modulates tomato thermotolerance and fruit quality, correlating with changes i-n autophage and heat shock-related genes.

Reply: Our sincere appreciation goes to the reviewer for the insightful comment and for reco-

mending a more accurate title. We acknowledge that our data illustrate a correlation rather thandefinitive causation. As a result, we have decided to adopt the reviewer’s proposed title verbati-m: “SlATG8f modulates tomato thermotolerance and fruit quality, correlating with variations in

autophagy and heat shock-related genes”. Please refer to the revised page 1 for details.

Comments: (2) The current structure is a disjointed list. Reorganize into clear subheadings to

improve readability.

Reply: We appreciate the reviewer’s valuable suggestion regarding the need for a clearer structure, as it enhances readability. Accordingly, we have meticulously restructured the “Materials and Methods” section by incorporating descriptive subheadings to establish a more cohesive and logical progression. Please refer to the revised page 4-8 for details.

Plant Materials and Growth Conditions

Generation of Transgentic Tomato Lines

Heat Stress Treatment and Sample Collection

Phenotypic Assessment of Fruit Development

Analysis of Fruit Quality Parameters

Measurement of Antioxidant Enzyme Activities and Malondialdehyde (MDA) Content

RNA Extraction and Gene Expression Analysis (qRT-PCR)

Statistical Analysis

We are grateful for your valuable guidance and await your feedback on our revised manuscript.

Comments: (3) Ensure consistent formatting for gene names. Some are Italicized,others are not.

Reply: Our sincere thanks to the reviewer for the detailed observation. We regret the error in

gene name formatting and have now meticulously reviewed the entire manuscript to ensure theconsistent italicization of all gene names (e.g., SlATG8f, ATG8, SlHSP90) throughout the text, f-igures, and figure legends, in compliance with standard scientific nomenclature. Protein names continue to be presented in nonitalicized text.

Comments:�4Please unify “SlATG8f-overexpressing (OE)” and “SlATG8f OE”by using the corre-ct form “SlATG8f OE plants”.

Reply: We thank the reviewer for this helpful comment. We have now carefully revised the en-tire manuscript to consistently use the unified term “SlATG8f OE plants” (or “SlATG8f-OE fru-its” where contextually appropriate) to describe the transgenic lines, as suggested.

We believe this change improves the clarity and consistency of the text.

Comments: (5)“Random sampling was performed” is vague, describe how the samples were selected and pooled if applicable.

Reply: We appreciate the reviewer’s insightful comment and acknowledge the need for a moreprecise description of the sampling method. Accordingly, we have enhanced the “Heat Stress T-reatment and Sample Collection” subsection in the “Materials and Methods” section to offer ac-omprehensive overview of our sampling procedure.explicitly outlines the random selection of fr-uits from various positi-ons on multiple plants for each biological replicate, and elucidates theprocess of pooling tissu-e samples for analysis. Please refer to the revised page 4-5 for detai-ls.

Comments: (6) “conventional soilless substrate” is not specific. The number of fruits sampled per stage, per plant, per treatment is not stated. A concise summary is required.

Reply: We appreciate the reviewer's constructive suggestions and have accordingly revised the manuscript to incorporate the specific details as advised.

In the “Plant Materials and Growth Conditions” section, the soilless substrate composition is detailed as follows: Seedlings were grown in a soilless substrate consisting of peat, vermiculite, and perlite in a volume ratio of 2:1:1.

In the “Heat Stress Treatment and Sample Collection” subsection, the sampling strategy is clearly outlined: “For each of the four developmental stages, nine fruits (three from each of three randomly selected plants) were collected per genotype per treatment to serve as biological replicates for further analyses.”

We are grateful for your valuable guidance and await your feedback on our revised manuscript.

Comments: (7)Correct the numerous typographical and spacing errors.

Reply: We express our gratitude to the reviewer for their meticulous observation. Following thi-s feedback, we conducted a thorough proofreading of the manuscript, rectifying all typographic-al and spacing errors to align with the journal's formatting guidelines.

We are grateful for your valuable guidance and await your feedback on our revised manus-cript.

Reply to the comments of Reviewer #2

Comments: (1)Title: the manuscript does not analyze the thermotolerance, I suggest remove thermotolerance from the title. Furthermore, HS should write the full name.

Reply: We thank the reviewer for bringing up these important points. In response to the suggestion regarding the term “thermotolerance,” we have carefully considered it. In the field of plant stress physiology, the capacity of plants to maintain functionality under high-temperature stress is viewed as an active response, demonstrating thermotolerance throughout their growth and development[1,2]. However, to accurately represent our correlational data�we have replaced the term “enhances”with “modulates” and presented the finding as a correlation. We believe that the revised title now effectively reflects our findings without excessive interpretation. Concerning the abbreviation “HS” we concur and have expanded it in the revised title to "heat shock.

We are grateful for your valuable guidance and await your feedback on our revised manus-cript.

Comments: (2)Key words: “High temperature stress” changed to“ high-temperature stress”.

Reply: We sincerely thank the reviewer for this careful correction. We have revised the key words from “High temperature stress” to “high-temperature stress” as suggested.

Comments: (3)Introduction: the role of autophagy in fruit development and quality need descripts.

Reply: We appreciate the valuable suggestion and acknowledge the importance of including background information on the role of autophagy in fruit development and quality in the introduction. Consequently, we have incorporated a new paragraph in the Introduction section to address this aspect. Please refer to the revised page 3 for details.

We thank you once again for this insightful recommendation.

Comments: (4)Many gene names should be in Italic, please check and modified. For instance, ATG in page 2.

Reply: Thank you for highlighting the importance of gene name formatting for standardization and professionalism in the paper. We apologize for any formatting errors in the original text. F-ollowing your guidance, we have meticulously reviewed and corrected the entire manuscript inaccordance with the journal's author guidelines. All gene names, including members of the AT-G gene family (e.g., ATG5, ATG7, ATG8a), the core gene SlATG8f studied here, and related g-enes, have been consistently italicized. This adjustment has been applied to the abstract, introd-uction, results, discussion, and figure legends throughout the document.

We appreciate your thorough review, which has significantly enhanced the quality of our s-ubmission.

Comments: (5)Abbreviations are defined upon their first appearance; subsequent occurrences require only the abbreviation itself, without repeating the definition.

Reply: We appreciate the reviewer for highlighting this oversight and apologize for any inconsi-stency in abbreviation usage. Following a thorough review, we have ensured that each abbrevi-ation is defined only upon its initial mention in the abstract, main text, and figure legends. S-ubsequent references now utilize the abbreviation alone. For instance, “heat stress (HS)”is defin-ed once, and “quantitative reverse transcription PCR (qRT-PCR)” is not reiterated in the Meth-ods section.

We are grateful for your valuable guidance.

Comments: (6)“L” in the Latin name for tomato should not be italic.

Reply: We express our gratitude to the reviewer for highlighting the significant technical detailconcerning the taxonomic nomenclature. The Latin name for tomato has been rectified througho-ut the manuscript to adhere to the correct format, “Solanum lycopersicum L”. Furthermore, wehave meticulously reviewed all scientific names in the text to ensure consistency with this stan-dard. We value your thoroughness in enhancing the accuracy of our manuscript.

Comments: (7)Delete “novel” in page 3.

Reply: The reviewer’s specific recommendation to remove the term “novel”as instructed on pag-e 3, is duly acknowledged. A comprehensive review of the manuscript has been carried out t-ensure the removal of similar subjective assertions. Emphasizing the importance of letting the f-indings speak for themselves,

we are grateful for this adjustment to enhance the objectivity of the language.

Comments: (8)M & M: F2 should be subscript of 2.

Reply: We sincerely thank the reviewer for highlighting this formatting error. We have now co-rrected “F2” to “F₂” in the “Plant Materials and Growth Conditions”section of the Materials and Methods. now consistently use the correct subscript format. We appreciate your attention to detail in helping us improve the manuscript's presentation.

Comments:�9�Agrobacterium and its Latin name should be italic.

Reply:We are grateful to the reviewer for identifying the oversight in the microbial nomenclature formatting. T

---

## [Editor Report · Decision Letter 1]

25 Nov 2025

Dear Dr. Xu,

Thank you for submitting your manuscript to PLOS ONE. After careful consideration, we feel that it has merit but does not fully meet PLOS ONE’s publication criteria as it currently stands. Therefore, we invite you to submit a revised version of the manuscript that addresses the points raised during the review process.

We look forward to receiving your revised manuscript.

Kind regards,

Timothy P. Devarenne, Ph.D.

Academic Editor

PLOS ONE

**Journal Requirements:**

**Additional Editor Comments:**

The edits you made to the manuscript are satisfactory. However, I request that you correct the word autophage in the title of your manuscript to autophagy. Please make this correction and resubmit.

---

## [Author Response · Author response to Decision Letter 2]

14 Dec 2025

Manuscript ID:PONE-D-25-51450R1

Title: SlATG8f modulates tomato thermotolerance and fruit quality, correlating with changes in autophagy and heat shock-related genes.

Author name:Qunmei Cheng;Wen Xu

Reply to the comments

Dear Editors:

We express our gratitude to the editorial office for the invaluable guidance provided during this second round of revision. All the points raised in your latest editorial comments have been thoroughly addressed, as outlined below. Your ongoing support is greatly appreciated, and we eagerly anticipate your final evaluation of our revised manuscript.

Reply to the comments of Editors:

Comments: (1) If the reviewer comments include a recommendation to cite specific previously published works, please review and evaluate these publications to determine whether they are relevant and should be cited. There is no requirement to cite these works unless the editor has indicated otherwise. 

Reply:We express our gratitude to the editorial office for the reminder regarding potential citati-on recommendations. Upon thorough re-evaluation of all reviewer comments, we confirm that no specific suggestions were made to cite additional publications. If the editor deems any parti-cular references necessary for the context of our study, we are willing to incorporate them as advised. Your guidance is highly valued, and we eagerly await your further evaluation of our revised manuscript.

Comments: (2)Please review your reference list to ensure that it is complete and correct. If yo-u have cited papers that have been retracted, please include the rationale for doing so in the manuscript text, or remove these references and replace them with relevant current references. Any changes to the reference list should be mentioned in the rebuttal letter that accompanies y-our revised manuscript. If you need to cite a retracted article, indicate the article’s retracted st-atus in the References list and also include a citation and full reference for the retraction noti-ce.

Reply: We express our gratitude to the editorial office for the reminder to ensure the com-pleteness and accuracy of the reference list. All references have been meticulously reviewed a-nd their formatting updated to adhere fully to the journal’s style guidelines. There are no citat-ions of retracted articles in this manuscript:

In response to previous review feedback, we have enhanced the manuscript by incorporating the following reference updates, all highlighted in red in the tracked-changes version:

Added a supporting reference in the Introduction to substantiate the statement: “While auto-phagy in crops has been extensively studied in contexts such as leaf senescence, seed development, and vascular formation”.

Included several pertinent references in the newly integrated section discussing the role of autophagy in fruit development and quality.

After a comprehensive assessment, I have determined that there is no need to cite a retrac-ted article. Therefore, I have not indicated the retraction status of the article in the reference list, nor have I cited the retraction notice separately or included the complete reference.

We affirm that the reference list is now precise, comprehensive, and free of retracted publ-ications. We value your guidance and anticipate your evaluation of our revised manuscript.

Comments: (3)The edits you made to the manuscript are satisfactory. However, I request that you correct the word autophage in the title of your manuscript to autophagy. Please make thiscorrection and resubm.

Reply: We are grateful to the editors for their favorable assessment of our revisions and for identifying the typographical error in the manuscript title. The correction from “autophage” to “autophagy” has been made as per their suggestion. We value the reviewer's meticulous review and await their final evaluation of our manuscript.

We express our deepest gratitude to the The manuscript has been revised to fulfill all editorial requirements. We appreciate your guidance throughout the review process. We are confident that these revisions have enhanced the paper and trust that it now meets the publication standards of PLOS ONE.

Yours sincerely

Wen Xu

December 14, 2025

---

## [Editor Report · Decision Letter 2]

16 Dec 2025

SlATG8f modulates tomato thermotolerance and fruit quality, correlating wi-th changes in autophagy and heat shock-related genes.

PONE-D-25-51450R2

Dear Dr. Xu,

We’re pleased to inform you that your manuscript has been judged scientifically suitable for publication and will be formally accepted for publication once it meets all outstanding technical requirements.

Kind regards,

Timothy P. Devarenne, Ph.D.

Academic Editor

PLOS One
---

## [Editor Report · Acceptance letter]

PONE-D-25-51450R2

PLOS One

Dear Dr. Xu,

I'm pleased to inform you that your manuscript has been deemed suitable for publication in PLOS One. Congratulations! Your manuscript is now being handed over to our production team.

Kind regards,

on behalf of

Dr. Timothy P. Devarenne

Academic Editor

PLOS One